# Glioblastoma Stem Cells—Useful Tools in the Battle against Cancer

**DOI:** 10.3390/ijms23094602

**Published:** 2022-04-21

**Authors:** Silvia Mara Baez Rodriguez, Georgiana-Adeline Staicu, Ani-Simona Sevastre, Carina Baloi, Vasile Ciubotaru, Anica Dricu, Ligia Gabriela Tataranu

**Affiliations:** 1Neurosurgical Department, Clinical Hospital “Bagdasar-Arseni”, 041915 Bucharest, Romania; mara.silvia@icloud.com (S.M.B.R.); dr_vghciubotaru@yahoo.com (V.C.); ttranu@gmail.com (L.G.T.); 2Department of Biochemistry, Faculty of Medicine, University of Medicine and Pharmacy, 200349 Craiova, Romania; adstaicu@gmail.com (G.-A.S.); carina_baloi@yahoo.com (C.B.); 3Department of Pharmaceutical Technology, Faculty of Pharmacy, University of Medicine and Pharmacy, 200349 Craiova, Romania; anifetea_umf@yahoo.com; 4Department 6—Clinical Neurosciences, “Carol Davila” University of Medicine and Pharmacy, 020021 Bucharest, Romania

**Keywords:** glioblastoma, stem cell, growth factor, signaling pathway

## Abstract

Glioblastoma stem cells (GSCs) are cells with a self-renewal ability and capacity to initiate tumors upon serial transplantation that have been linked to tumor cell heterogeneity. Most standard treatments fail to completely eradicate GSCs, causing the recurrence of the disease. GSCs could represent one reason for the low efficacy of cancer therapy and for the short relapse time. Nonetheless, experimental data suggest that the presence of therapy-resistant GSCs could explain tumor recurrence. Therefore, to effectively target GSCs, a comprehensive understanding of their biology and the survival and developing mechanisms during treatment is mandatory. This review provides an overview of the molecular features, microenvironment, detection, and targeting strategies of GSCs, an essential information required for an efficient therapy. Despite the outstanding results in oncology, researchers are still developing novel strategies, of which one could be targeting the GSCs present in the hypoxic regions and invasive edge of the glioblastoma.

## 1. Introduction

Glioblastoma (GBM) is the most common type of primary malignant brain tumor, accounting for 55% of primary brain tumors. GBM has a poor prognosis, with a mean survival rate between 14 to 16 months for patients treated with the standard treatment strategies (maximal safe surgical resection, followed by radiotherapy together with concomitant and adjuvant chemotherapy using temozolomide (TMZ)). Temozolomide is the first-line chemotherapy drug in GBM, an available orally alkylating agent that causes apoptosis by generating single-strand and double-strand breaks in DNA [1]. The 2-year survival rate was still less than 30%, even in the most recent reports.

GBM is a heterogeneous tumor resistant to all therapeutic approaches because of the multiple subclonal driver mutations [2,3]. Almost all tumors consist of various cell populations and heterogeneity that make them difficult to treat. The causes of recurrence are complex and include the absence of a clear tumor margin useful for a complete resection, the presence of migrating cancer cells into the surrounding normal tissue, high proliferative index, chemotherapy and radiotherapy resistance of the cancer stem cells (CSCs), and cerebrospinal fluid (CSF) dissemination. CSCs, also known as tumor-initiating cells (TICs), are suggested to be responsible for cancer relapse and drug resistance due to their ability to self-renew and to differentiate into a heterogeneous population of cancer cells [4,5]. 

Therefore, the surgical removal of the entire tumor mass is impossible. The presence of tumor cells at 2–3 cm from the original site of the tumor has led to even more aggressive tumoral recurrences after the removal of the tumor mass [6,7]. 

TICs and GSCs represent a subpopulation of self-renewing cells involved in the process of tumor initiation (for TICs) and tumor maintenance (for GSCs). Both types have distinct markers and biological functions and can be approached using different molecular and cellular methods. GSCs may be derived from TICs [8].

TICs (referred to as the cell of origin) are normal cells that acquire the first mutation that results in cancer promotion. The cells that promote GBM regrowth after surgical resection are supposed to be the undifferentiated GSCs, which produce differentiated progeny, creating a rapidly dividing tumor. They are also responsible for tumor heterogeneity and therapy resistance [9,10,11,12]. These cells demonstrate the main characteristics of stem cells: proliferation, differentiation, and self-renewal [9,13,14].

The GSCs’ resistance to radio- and chemotherapy could be explained by their high capacity for extensive DNA repair, quiescence, higher mitochondrial reserve, and localization in the hypoxic niche [9]. Mitochondrial activity is modulated by hypoxic conditions, controlling cell apoptosis and necroptosis, and reactive oxygen species (ROS) generation, therefore reducing susceptibility to chemo- and radio-induced apoptosis [15,16]. Moreover, anticancer therapies induce cell death by direct or indirect DNA damage. Targeting the DNA repairing pathways could increase the sensitivity of tumor to various cancer therapies [17]. Quiescence is a mechanism that turns the stem cells into a low metabolic state. The quiescence is correlated with the tumor microenvironment, including the extracellular matrix, immune cells, signaling molecules, and surrounding blood vessels. In GSCs, low ROS levels are associated with the quiescence/dormancy state of stem cells and with a protective intracellular environment [18].

According to the 2021 World Health Organization (WHO) Classification of Tumors of the Central Nervous System (CNS), GBMs are classified as “adult-type diffuse gliomas” (a grade IV tumor) [19]. The presence of extensive microvascular hyperplasia, hypercellularity, proliferation, necrotic foci surrounded by pseudopalisading cells and diffuse infiltrative margins differentiates them from low-grade gliomas [20]. 

Additionally, immunohistochemistry (IHC) is considered a fast and reliable substitute for molecular genetic tests in diagnostic surgical pathology. A narrow panel of highly sensitive immunohistochemical biomarkers can predict with a high accuracy the GBM type. For this matter, based on the IHC expression of four proteins (CD44, p53, IDH1, and PDGFRA), the study on gliomas published by Jakovlevs et al. indicated the presence of two mutually exclusive molecular signatures similar to proneural and mesenchymal subtypes [21].

Furthermore, integrated algorithms based on IHC can offer high accuracy in predicting glioblastoma transcriptional subtypes. In a study published by Orzan et al., mesenchymal and classical subgroups proved to be well segregated, and the proneural types showed a mixed proneural/classical phenotype, predicted by the algorithm as proneural, but with comparable probability to be a part of a classical subtype [22].

The tissue is protected from harmful molecules by the existence of the blood–brain barrier (BBB) [23]. This is formed by astrocytes and pericytes that surround the endothelial cell tight junctions [24]. Due to the existence of this barrier, the delivery of effective drug concentration is sometimes difficult, as it regulates the extravasation of macromolecules and chemotherapy drugs [24]. Because of the high genetic heterogeneity [25,26,27], it is impossible to target all GBM cells using only one biomarker-targeted therapy. Most of the GSCs are found in the hypoxic and necrotic areas, which are difficult to penetrate by the chemotherapy drugs. Furthermore, their resistance to chemotherapy is facilitated by various mechanisms, such as drug metabolic inactivation, decreased drug influx, overexpression of drug efflux pumps, slow division rate, inhibition of prodrug to bioactive drug conversion, and increased double-strand DNA repair [9,28,29,30]. The mechanism of the drug efflux is energy dependent, and it is facilitated by the increased expression of the ATP-binding cassette superfamily (ABC) of transporters [31], which are overexpressed in GSCs [32]. 

Aldehyde dehydrogenase 1 (ALDH), an enzyme that detoxifies alkylating agents, reducing their reactivity, and O^6^-methylguanine-DNA methyltransferase (MGMT), a detoxifier enzyme, are both contributors to GBM chemoresistance and highly expressed by GSCs [33,34,35].

ALDHs are markers of CSCs indicating worse prognosis in GBM [36]. ALDHs have not been proved to be linked to DNA repairing pathways, and the mechanism of how they mediate chemoresistance remains unclear. Clinical data reported increased levels of ALDH1A3 in recurrent GBM tumors [35]. It has been proved that ALDH1A3 is involved in ROS reaction. ROS react with polyunsaturated fatty acids from lipid membranes, inducing lipid peroxidation. ALDH1A3 seems to reduce the extent of toxic aldehydes resulting from lipid peroxidation [37].

Nevertheless, MGMT is directly responsible for the repair of lesion. Epigenetic silencing of the MGMT gene inhibits the synthesis of MGMT, increasing the sensitivity to cytotoxic effects induced by alkylating compounds. MGMT methylation is a predictor of longer survival for diagnosis but not for recurrence, suggesting that other mechanisms are responsible for MGMT upregulation in recurrent tumors [38]. Metastatic mismatch repair gene (MMR) alterations have been described in 10% to 20% of recurrent tumors, but changes in its promoter methylation status have been detected in a few patients [3]. It has been suggested that enhancer hijacking in recurrent GBMs could promote the MGMT expression and, therefore, alkylating compounds’ resistance, but this clinical significance remains to be evaluated [38,39].

Nonetheless, it has been demonstrated that radiotherapy is also insufficiently effective in destroying the GSCs [40,41]. Because of their high resistance to drugs, radiation, and surgery, the GSCs represent an important therapeutic target, intensively studied worldwide in the last years [41].

## 2. GSCs’ Biomarkers

In order to achieve an optimum treatment efficacy, it is important to identify the GSCs from the rest of the tumoral cells. Glioblastoma stem cells present several biomarkers used for identification, such as CD133, nestin, musashi-1, CXCR4, CD15, CD34, CD44, SOX2, L1CAM, and A2B5, however, neither being exclusively characteristic for GSCs [42,43,44,45,46,47,48,49,50,51,52,53,54] (Table 1). Furthermore, this task becomes even more difficult because GBMs have the ability to remodel their microenvironment by modulating the immune system, vasculature, and stroma [44].

## 3. GSCs and Tumor Microenvironment

Even though several hypotheses have been proposed, none can fully explain the origin of the GBM. These hypotheses are based on the dedifferentiation of neural cells, transformation of undifferentiated precursor cells, and proliferation of neural stem cells [55,56,57]. Campos et al. stated that dedifferentiation may occur when an accumulation of genetic mutations in oncogenes appears in normal brain cells [58]. Additionally, genetic mutations in neural stem cells may cause the formation of cancer cells. 

It seems that GSCs increase tumorigenesis by recapitulating the normal neural lineage hierarchies of quiescence and self-renewal. By maintaining stemness in specialized niches and by directing differentiation on the appropriate lineages, the microenvironment is controlling the normal neural stem cells’ fate. While GSCs are found in hypoxic and perivascular regions of the tumor bulk, not much is known about the GSCs’ differentiation potential within tumors and neither whether a prodifferentiative niche exists [59]. 

GBM may develop in the white matter and spread inside the brain via myelinated fibers, but the cellular and molecular processes that support the white matter invasion remain unknown. However, there are studies that revealed that the white matter may suppress malignancy by directing the differentiation of GSCs towards preoligodendrocyte fate [60]. 

Neural stem cells have two places of origin (also called neurogenic niches): the subventricular zone, located in the forebrain lateral ventricle, and the subgranular zone, located in the hippocampus in the dentate gyrus. Stem cells in quiescent or active mitotic state can be found in both regions [61].

Three major microenvironments have been described in glioblastoma: the hypoxic niche (around the necrotic core), the perivascular niche, and the invasive edge [62].

### 3.1. The Perivascular Niche

Vasculogenesis is the process of de novo formation of blood vessels that occurs mostly during fetal development. Studies showed that circulating endothelial cells, tumor-associated macrophages (TAMs), Tie-2 monocytes, myeloid-derived suppressor cells (MDSCs), neutrophiles, and GSCs are involved in the formation of new vessels [63,64,65,66]. GBMs are characterized by increased angiogenesis (process of stimulating the formation of new vessels from pre-existing ones) [2,4,5,9,13].

The perivascular niche is formed by nonmalignant cells (astrocytes, fibroblasts, pericytes, immune cells, neural progenitor cells) and malignant cells (GSCs and tumor cells located around disorganized blood vessels) [62]. These cells interact between them, supporting GSCs’ survival and growth. Under hypoxic conditions, the tumor produces angiogenic factors involved in the formation of the tumoral blood vessels. 

More than 2 decades ago, the glioma proliferation was linked to growth factors and their receptors [67,68]. These factors include epidermal growth factor (EGF), vascular endothelial growth factor (VEGF), fibroblast growth factor (FGF), platelet-derived growth factor (PDGF), mammalian target of rapamycin (mTOR), protein kinase C (PKC), histone deacetylase, farnesyltransferase, heat shock protein 90 (Hsp90), and histone deacetylase [69]. They are important mediators of angiogenesis, affecting the oxygen supply to tumors [70,71]. 

A variety of inhibitors became available (Table 2), and their effectiveness was demonstrated on glioblastoma using in vitro and in vivo studies [72,73,74,75,76,77,78,79,80,81,82]. 

Furthermore, GBM presents abnormal and fragile blood vessels. The rupture of the vessels leads to the disruption of the BBB [58]. These vessels are the result of the VEGF involvement in pericyte disintegration. After BBB disruption, tumor-derived chemokines attract immunomodulatory cells that enter the brain. They increase the angiogenic factors’ production, suppressing the immune function and leading to tumor progression [62]. Vascular pericytes attach to endothelial cells and play an important role in maintaining the BBB. The depletion of pericyte may disrupt the BBB and may elevate the vascular permeability [83]. In GBM, most of vascular pericytes derive from GSCs via transdifferentiation. They express tumor-specific genetic alterations, differentiating them from normal pericytes; therefore, the neovasculature and tumor growth may be potently inhibited by selective elimination of these GSC-derived pericytes. GSCs are recruited through the SDF-1/CXCR4 axis and are further induced to transform into pericytes by transforming growth factor β (TGF-β). However, GSCs may actively remodel perivascular niches by their contribution to vascular pericytes [84]. 

Different immune-suppressive mechanisms are used in GBM to prevent its immune detection and eradication. GBM cells secrete a variety of immunosuppressive proteins. Intracellular adhesion molecule 1 (ICAM-1) is a cell–cell interaction key regulator that is commonly upregulated in GBM. ICAM-1 promotes the migration of myeloid cells into tumors by interacting with lymphocyte function-associated antigen 1 (LFA-1) expressed on these cells. The accumulation of MDSC in GBM further contributes to immune suppression. Additionally, GBM overexpresses galectin-1 (Gal-1), which promotes tumor cell proliferation and migration [85]. 

Furthermore, regulatory T cells (Tregs) that either can originate in the thymus (naturally occurring Tregs) or can be induced by antigens (iTregs) could contribute to GBM-mediated immune suppression. Tregs suppress the immune responses by cytokine secretion, such as interleukin (IL)-10 and TGF-β, or by cell-to-cell mediated contact. Cytotoxic T-lymphocyte-associated protein 4 (CTLA-4) is present on the surface of activated T effector cells. Together with CTLA-4, programmed death 1 (PD-1) is an immune checkpoint that forms a system that regulates the immune activation and proliferation. Studies of tumor microenvironments revealed that this system is capable of inducing T-cell apoptosis [86,87,88].

More recently, an orphan member of the adhesion G-protein-coupled receptor family, EGF, latrophilin, and seven transmembrane domain-containing protein 1 (ELTD1) was reported to be upregulated in high-grade glioma blood vessels, and its expression has been associated with glioma progression and has also been suggested as a potential therapeutic target in glioblastoma [89,90,91,92]. 

TAMs play a significant role in the perivascular niche. TAMs are attracted in the niche by GSCs’ secreted periostin [93]. They produce heme-oxygenase 1 (HO-1) and thymidine phosphorylase (TP) involved in neovascularization. They express chemoattractants, such as VEGF, interleukin (IL)-6, colony-stimulating factor (CSF), stromal cell-derived factor 1α (SDF-1α), and IL-1β. The attraction of macrophages and monocytes generates an immunosuppressive phenotype and tumor progression. Additionally, TAMs produce TGF-β involved in matrix metalloproteinase 9 (MMP9) expression. This process causes further GSC proliferation [94]. 

The vasculature and angiogenic factors play a pivotal role in glioblastoma induction and the maintenance of immunosuppression. Proinflammatory repolarization of macrophages in the perivascular and perinecrotic tumor is probably an option to overcome treatment resistance [95].

Similar to VEGF and CSF, the CD34 role has been discussed in the diagnosis of various cancer pathologies. It has been demonstrated that it is involved in promoting a new blood vessel network and in increasing the nutrients and oxygen supply for further tumor growth [96]. In GBM, CD34 has been found to be expressed in the tumor vascular endothelial cell membrane. In gliomas, the CD34-positive cells are recruited by bone-marrow-derived circulating hematopoietic progenitors. In our recent study, we found no correlation between different grades of glioma or tumor vascularization and CD34 expression and with mild distribution of CD34 in CNS tumors [97]. Four types of CD34-labeled microvessel formations in glioblastomas have been detected based on different vascular niche pathologic structures: microvascular sprouting, vascular cluster, vascular garland, and glomeruloid vascular proliferation [98]. 

Targeting the perivascular niche could represent an effective approach for GBM therapy by using angiogenic factors’ inhibitors to influence the tumor aberrant vascular proliferation. To inhibit GSC differentiation, this strategy may be combined with other treatments, such as repolarizing macrophage-designed immunotherapeutics [99].

### 3.2. The Perinecrotic or Hypoxic Niche

Hypoxia is one of the main characteristics of glioblastoma. The insufficient blood supply causes hypoxia, leading to pseudo-palisading necrosis. Normal tissular median oxygen saturation is approximately 5%. Instead, in the necrotic regions, the oxygen concentration is less than 2%. This is determined by higher metabolic activity and increased oxygen consumption in the heterogeneous tumor cells [100]. 

Located around the necrotic core, the hypoxic niche is proved to be involved in tumor growth, cell maintenance, stemness induction, and immune surveillance [101]. The low oxygen concentration upregulates important proteins, such as hypoxia-inducible factors (HIF1 and HIF2), a dimeric protein complex with an important role in the angiogenesis and dedifferentiation process [55,56,57]. It has been found that many GSCs reside in this niche [62,97,101]. Hypoxia can induce stemness characteristics and determine the increased expression of GSC markers [20,29,33]. GSCs and tumor cells may survive in the hypoxic niche after chemo- and radiotherapy. The cellular death in the necrotic area generates proinflammatory signals, converting inflammatory cells into immunosuppressive cells and inducing angiogenesis [102]. Hypoxia can cause the differentiation of GSCs into endothelial cells, promoting the tumor growth from the necrotic area towards the neovascular region [103].

Hypoxia plays a pivotal role in the SRC tyrosine kinase pathway. The increase in SRC activity upregulates the VEGF in low-oxygen conditions. Moreover, a correlation between angiogenesis and hypoxia has been proved by the increased vascularization, with integrin upregulation. Additionally, it has been demonstrated that cell survival mechanisms are activated as a result of increased SRC signaling in response to an anti-VEGF agent [104]. In addition, the hypoxic conditions increase glycolysis because of the MYC oncoprotein. Its regulation has been linked to the SRC pathway in other tumors [105]. Therefore, the SRC–MYC axis may be implicated in metabolic reprogramming also in GBM, apart from the receptor tyrosine kinases’ (RTKs) involvement. The hypoxia-induced SRC pathway results in fostered invasiveness. It consists in integrin β3 and EGFR-vIII interaction, αvβ3 integrin recruitment on cell membranes, and FAK activation, which creates focal adhesion complexes. Furthermore, the EGFRvIII/integrin β3/FAK/SRC axis continues with the intracellular signaling pathway ERK1/2, MAPK, AKT, and STAT3 activation, determining the MMP upregulation, therefore promoting cell invasion [106].

In a recent study, Torrisi et al. evaluated the relationship between hypoxia and radiation-induced radioresistance with the activation of SRC proto-oncogene nonreceptor tyrosine kinase, prospecting potential strategies to overcome the current limitation in glioblastoma treatment. Some promising ways to sensitize GBM cells could be represented by “hypoxia dose painting” (personalized radiation dose as a function of local microenvironmental or phenotypic variations), combination of multiple ion beams in order to deliver high-linear energy transfer radiation in the hypoxic area, reoxygenation strategies, and targeting the molecular mechanisms involved in hypoxia [107]. 

### 3.3. The Invasive Niche

GBM tumor cells can migrate along normal blood vessels, invading normal brain tissue [5]. The deletion of MMP2 and MMP9 boosts the perivascular invasiveness and diminishes angiogenesis [108]. The most important cell of the invasive niche is the astrocyte. The close contact between the astrocytes, endothelial cells, and pericytes facilitates the transport of ions and metabolites from the blood vessels to the brain [97]. This microenvironment is heterogeneous, and it seems that cell populations within it are plastic. At the cellular level, malignant cells invade diffusely inside the surrounding normal brain parenchyma, followed by increased proliferation of endothelial cells, generating leaky blood vessels and resulting in a hypoxic microenvironment. The invading GBM cells displace the pre-existing astrocytes and pericytes, thereby disrupting the BBB and resulting in leaky blood vessels [109]. Paracrine interaction at this level can cause astrocyte proliferation and migration. Among the many receptors expressed by the GSCs, integrin-1 and tropomyosin receptor kinase type A (TRKA) can bind to the connective tissue growth factor (CTGF) released by the astrocytes, and produce zinc finger E-box binding homeobox-1 (ZEB1), leading to tumor cell infiltration [110,111].

It has been discovered that astrocytes express Sonic hedgehog (SHH), important glioma-associated oncogene (GLI) activator, and stemness promoter [112]. Therefore, astrocytes play a key role in GSCs’ preservation and in tumor spreading. Thus, it is expected that large tumors have extensive invasive niches, intense hypoxia, and increased neoangiogenesis.

## 4. Strategies for Targeting GSCs

The GSCs could be targeted directly by blocking their signaling pathways, by promoting GSC differentiation, and by virotherapy, or indirectly by targeting the perivascular, hypoxic, or immune GBMs niche. 

Therapeutic targeting of GSC pathways and their receptors implicated in cell proliferation, maintenance, and tumor resistance is of great interest by providing reliable trials to block them. Among these, tyrosine kinase inhibitors (TKIs) have proved less advantages compared with GSC pathways (Notch, Wnt, and SHH) [99] 

Notch, SHH, and Wnt are signaling pathways that regulate the cellular processes, including the differentiation, proliferation, and migration of GSCs (Figure 1).

The Notch pathway (Notch 1–Notch 4 receptors and Jagged-1 and Jagged-2 and delta-like-1, delta-like-3, and delta-like-4 ligands), can modulate the interaction between neighbor cells. This pathway is activated by proteolytic cleavage reactions, and inhibitors of the γ-secretase complex play a pivotal role in blocking the Notch signaling pathway [113].

Additionally, the Wnt/β-catenin pathway is important in modulating the self-renewal and differentiation of GSCs. The nuclear stabilized β-catenin abnormally activates the β-catenin pathway; therefore, it can be correlated to tumorigenesis, growth invasion, and MGMT expression, incriminated for TMZ resistance [114]. 

Furthermore, the activation of the SHH signaling pathway upregulates the multi-drug-resistance-associated protein-1(ABCC1/MRP1), ABC transporter ABCG2, drug efflux P-glycoprotein (ABCB1), B-cell-specific Moloney murine leukemia virus integration site 1 (BMI1), and MGMT. Additionally, the loss of p53 upregulates the transcription factor Nanog correlated with the SHH signaling pathway, probably contributing to TMZ chemoresistance by regulating the expression of MGMT [115].

RTKs are involved in GSC expansion and differentiation. The EGFR overexpression has been observed in 40–60% of primary human GBM tumors [115]. It is implicated in frame deletions in the extracellular domain and in gain-of-function missense mutations and less in the inhibitor responsiveness [116]. Three types of EGFR inhibitors are currently used in the cancer field: first-generation reversible small-molecule inhibitors that target EGFR and coreceptor HER2 (erlotinib and gefitinib), second-generation inhibitors that irreversibly bind to EGFR (afatinib, dacomitinib, and neratinib), and third-generation irreversible TKIs (AZD9291). Furthermore, in a recent study, SH3 domain containing kinase binding protein 1 (SH3KBP1), an activator and modulator of the EGFR signaling, has been proved to have increased levels of mRNA and protein in GBM. In addition, there is evidence that SH3KBP1 is predominantly expressed in GSCs, which makes it a promising novel GSC marker and a future target [117]. 

The PDGF receptors are responsible for developing oligodendrocytes and blood vessels during embryonic development. Overexpression and alterations of PDGF ligands are commonly present in GBM tumors, especially in the proneural subtype, contributing to GBM development and GSC functions. Several inhibitors (imatinib, tandutinib, nilotinib, and AG1433) demonstrated their efficacy to decrease the growth of GBM xenografts in vivo, but with no significant effect on recurrent GBM [118].

Furthermore, GBM highly expresses VEGF and its receptors, important regulators of angiogenesis, vasculogenesis, and lymphangiogenesis. There are many VEGFR inhibitors with various treatment effects, such as cediranib, lapatinib, pazopanib, sorafenib, atalanib, tivozanib, and SU1498 [87]. Recently, the human cartilage glycoprotein-39 or chitinase-like protein-1 (YKL-40) has been proved able to upregulate the expression of VEGF. YKL-40 is a molecular marker for a mesenchymal subtype of GBMs found to be responsible for TMZ, resistance that gives it a chance to become an effective target in cancer therapy [119].

Another strategy to directly target the GSCs is by promoting GSC differentiation. It seems that GSCs become more sensitive to various therapies after differentiating in more terminal GBM cells [120]. Therefore, a key in the therapy against GBM could be achieved by promoting the GSC differentiation. Several attempts have been made. For example, the bone morphogenic proteins are involved in the stem cell niche and in GSC differentiation [121]. Another example is the overexpression of miR-128, which influences differentiation and enhances the senescence mediated by axitinib [122]. Moreover, the graphene oxide decreases the expression of stem cell markers and increases the expression of differentiation-related markers [123]. Additionally, the nonsteroidal anti-inflammatory agent sulindac has been proved able to induce GSC differentiation [124].

Among various strategies that target GSCs, oncolytic virotherapy has been proved efficient in preventing GBM recurrence. For this purpose, the oncolytic adenovirus DNX-2401 (BM-hMSCs-DNX2401) has been administered with good results intra-arterially, loaded in allogeneic bone-marrow-derived human mesenchymal stem cells [125]. 

Instead, the indirect strategy of targeting the GSC niche involves targeting the perivascular niche through angiogenic pathways, the immune niche, and the hypoxic or perinecrotic niche by inducing hypoxia. 

The perivascular niche is surrounded by a vasculature with different characteristics compared with normal blood vessels, featuring structure and function abnormalities, hypoxia, and hierarchy impairment. There is a codependent and synergistic relationship between the perivascular niche of GBM and GSCs: the GSC survival, proliferation, and migration are maintained by GBM, which mediates the perivascular region, and GSCs simultaneously act as a cancer-specific vasculature regulator, infiltrating into the tumor [126]. GSCs express high levels of angiogenic factors, providing an optimum environment for GSC survival and proliferation; therefore, the blockage of angiogenesis in the perivascular niche could diminish the original tumor.

Moreover, in GBM therapy, the immune niche could also be targeted. Tumor aggression increased, and immune cells infiltrated in the brain after bevacizumab or cediranib treatment of recurrent GBM in GBM mouse models [127]. The resistance of antiangiogenic therapy could be explained by the direct interaction between GSCs and the immune cells, leading to VEGF-independent angiogenesis and generating tumors without the ability to respond to inhibitors [128]. By targeting the innate immune cells and receptors, tumor sensitivity increased, a strategy that could be a promising therapy for GBM.

Hypoxia is one diagnostic hallmark of GBM. Hypoxia-inducible factor-1α (HIF-1α) and VEGFA are two main regulators of hypoxia response, critically implicated in worse progression-free survival. Thus, bevacizumab, a VEGFA inhibitor, is a promising agent, useful in overcoming both hypoxia and resistance in cancer therapy [87,129]. 

## 5. GSCs Biology, Genetic and Epigenetic Changes

GBM features both genetic and epigenetic alterations responsible for cancer cell gene expression regulation. The epigenetic mechanisms cause gene expression alterations in response to the environment. Some of the epigenetic changes that occur in GSCs are:The DNA methylation, mostly within gene-promotor area (CpG) islands, which influence gene expression. Hypermethylation of tumor suppressor genes, such as TP53, generally causes tumorigenesis [130], while usually hypomethylation leads to the oncogene activation. Early findings showed that many tumor suppressor genes are targets for DNA hypermethylation in cancer, therefore the idea that aberrant DNA methylation may promote oncogenesis via tumor suppressor gene silencing. However, more recent genome-wide analyses have proved that the classical model requires to be reconsidered [131];Changes in the microRNA (miRNA) family, small single-stranded noncoding functional RNA molecules involved in RNA silencing and post-transcriptional regulation. Studies proved the existence of 351 altered expressions of miRNAs in GBM, 256 being overexpressed and 95 underexpressed, demonstrating that the miRNA expression is modified in GSCs compared with the normal brain cells [132];Dysregulation of a polycomb group of proteins (PcGs) causes tumor progression and invasion. They can determine gene silencing by remodeling chromatin. One of these proteins, BMI1, a member of PRC1 (polycomb repressive complex 1), blocks the differentiation of GSCs into neurons and inhibits apoptosis [133]. EZH2, a member of PRC2, is suspected to be involved in supporting the GSCs by activating STAT3 [134,135]. This protein group is considered to support GBM progression and invasiveness;Additionally, GBM formation explained by post-translational modification of histones [136], influencing the chromatin architecture, with epigenetic changes.

To efficiently target GSCs, the mechanisms behind the genetic and epigenetic alterations, the molecular pathways, and the interactions between the tumor microenvironment and GSCs must be thoroughly understood. 

Overall, a permanent epigenetic block supports the self-renewal ability of the GSCs, featuring them with the impairment of differentiation [137]. Still, GSCs can modulate their DNA methylomes and transcriptomes, rapidly adapting to various microenvironments, suggesting that these alterations may be partially reversible, differently from genetic variations. The epigenetic mark’ reversibility has been demonstrated in glioma cells by using a certain combination of transcription factors and by reversing in an early embryonic state, followed by a general resetting of DNA methylation associated with cancer. However, this resetting could not cancel the malignant behavior of the GSCs, proving that epigenetic-related activities need to be thoroughly deciphered in order to explain their malignancy [138,139].

The characteristics of GSCs follow regional variations according to the tumor niche. The regions with BBB disruption are characterized by increased expression of proneural genes (EZH2, SUZ12, H3K27me3), while the hypoxic necrotic regions have a high expression of mesenchymal genes and BMI1 targets, with a strong H2A119ub association, and the presence of CD44 and YKL40 markers. Nonetheless, selective inhibition of BMI1 or EZH2 was proved to be effective against the survival of both mesenchymal and proneural GSCs [139]. 

GSCs upregulate multiple receptors and signaling pathways that could represent the source of CNS oncogenic alterations. They have been found to be responsible for tumor proliferation, resistance to radiotherapy and chemotherapy, maintenance, and aberrant cell survival [40,57,138]. The signaling pathways and the corresponding receptors have a pivotal role for GSC involvement in oncology. Notch, SHH, and Wnt are the most representative signaling pathways, with large implications in the cancer domain.

Artificially induced replication stress in glioma cells has been associated with radio resistance. Replication stress is represented by an inefficient DNA replication that determines replication forks to evolve slowly or even to stop. Replication stress activates some molecular processes that stabilize the replication forks, preventing DNA damage. Replication stress is considered to be one mechanism for radiotherapy resistance glioma stem cells, proved by the elevated levels of markers, such as replication protein A, DNA damage markers, and single-stranded DNA binding protein [140]. Additionally, tyrosine kinase MET induces radio resistance via activation of AKT kinase and DNA repair downstream effectors and by phosphorylation of the p21 protein with an antiapoptotic effect [141]. Targeting replication stress response with combined ATR (ataxia telangiectasia and Rad3 related) and PARP (poly (ADP-ribose) polymerase) inhibition provides high and specific cytotoxicity in glioma cells. Combined ATR and PARP inhibition leads to a specific vulnerability, resulting in DNA low replication and abrogation of DNA repair [142]. 

### 5.1. Notch Signaling Pathway

The Notch signaling pathway is a highly preserved cell signaling system present in most animals. It is involved in cell fate, migration, proliferation, and cellular quiescence maintenance and is also a regulator of neural stem cell differentiation [139]. The multipotency perpetuation is stimulated by the activation of the HES (hairy/enhancer of split) and HEY (hairy/E(spl)-related with YRPW motif) genes and by the formation of an RBPJ (recombining binding protein suppressor of hairless) and MAML (mastermind-like) protein complex in the nucleus. This process is followed by the translocation of NICD (Notch intracellular domain) to the nucleolus caused by the cleavage of the Notch receptor by presenilin/α-secretase, after binding Jagged and delta Notch ligands [143,144]. The high expression of the Notch signaling pathway activators ID4 (inhibitor of differentiation 4) and FABP7 (fatty-acid-binding protein 7) by the CD133-positive GSCs influences the migration of radial ganglion cell (RGC) and stimulates the infiltrating capacity of GBM [145]. It is yet to be discovered how Notch is activating the GSCs’ oncogenic signaling mechanism. It is already known that Notch activation is amplified by tenascin-C, an extracellular matrix protein [146], and it is associated with GSCs’ stemness maintenance. This supports the idea that blocking the Notch signaling pathway may be an effective strategy of limiting tumor progression by GSC inactivation.

### 5.2. SHH/GLI Signaling Pathway

SHH has an important role in regulating embryonic morphogenesis. In adults, it is involved in tissue repair processes, with great implications in GSCs’ self-renewal mechanism and tumorigenicity [99,147]. Additionally, the SHH/GLI pathway is involved in upregulating proteins that are highly activated in most of the GSCs, such as drug efflux P-glycoprotein (ABCB1), breast cancer resistance protein (BCRP/ABCG2), multi-drug-resistance-associated protein 1 (MRP1/ABCC1), O^6^-methylguanine-methyltransferase (MGMT), and BMI1 [148,149]. Nanog, a protein-coding gene, considered to be the key to pluripotency, is one principal regulator of the expression of certain stemness factors. The Nanog expression is activated by binding SHH/GLI1/GLI2 to the Nanog promoter. Furthermore, p53 is involved in downregulating the Nanog expression due to the decrease in GLI1 expression and activity. Therefore, the loss of p53 may contribute to the maintenance of stemness by activating SHH signaling and by upregulating Nanog [150]. Studies have shown that targeting this pathway could increase the efficacy of chemotherapy [151].

In a study performed by Torrisi et al., SHH signaling and CX43-based intercellular communication has been modulated using in vitro models of proliferation and migration. They suggested a potential axis between the SHH signaling pathway and CX43 at least on two aspects: “permissive”, in which SHH-GLI signaling favors the intercellular communication and patterning, leading to microenvironmental changes and tumorigenic onset, and “supporting” the GBM stemness signature so that the aberrant SHH-GLI pathway promotes GSC population [152].

### 5.3. Wnt Signaling Pathway

The Wnt signaling pathway plays an important role during differentiation and self-renewal of the neural stem cells [153] as well as GSCs [154]. Its abnormal activation can lead to the formation of brain tumors. The Wnt pathway is involved in GSC maintenance by means of genetic and epigenetic mechanisms [154]. The overexpression of PLAGL2 (pleomorphic adenoma gene-like 2) increases the expression of various Wnt receptors, such as FZD2 (frizzled), FZD 9, and Wnt6 [155]. FoxM1, a proliferation-associated transcription factor involved in cell proliferation, self-renewal, and tumorigenesis, generates stemness and preservation of the GSCs after binding to Sox2 (SRY-box transcription factor 2) [156]. Evi (evenness interrupted), a transmembrane protein overexpressed in GBM, is regulated by epigenetic mechanism, modifying both canonical and noncanonical Wnt pathways [157]. Wnt5A causes endothelial differentiation of GSCs, leading to neovascularization, a favorable condition for cell growth and invasion [158]. 

Both canonical (characterized by the intracellular accumulation of β-catenin) and noncanonical (defined by its β-catenin-independent actions) Wnt signaling pathways play a significant role in gliomagenesis and the maintenance of GSCs. Furthermore, the TMZ resistance can be explained by the Wnt-induced expression of MGMT [159]. 

Because of its involvement in multiple essential functions in the entire body, the Wnt signaling pathway cannot be targeted without various side-effect consequences [153,160,161]. 

## 6. Conclusions

Although GBM is continuously being in the focus of researchers worldwide, the gathered information has not led to a major breakthrough in therapy. The standard treatment strategy continues to be the safe maximal surgical resection, followed by radiotherapy and chemotherapy. Unfortunately, these protocols are not able to safely cure cancer patients, but only to prolong their survival rates. Regularly, the treatment is followed by often more aggressive recurrences or metastasis, most probably because of the tumor microenvironment and molecular heterogeneity. 

In this review, we focused on GSCs, detailing their vital signal pathways, resistance mechanism, and crosstalk between them and their niche. We also provided different GSC-targeting strategies, a great opportunity for further approaches. Accordingly, more studies need to be performed to have a better framework regarding the properties, origin, and progression of GSCs, as well as their maintenance, resistance, and recurrence. In the future, novel approaches that target GSCs could become useful tools for novel clinical applications that somehow are urgently needed.

## Figures and Tables

**Figure 1 ijms-23-04602-f001:**
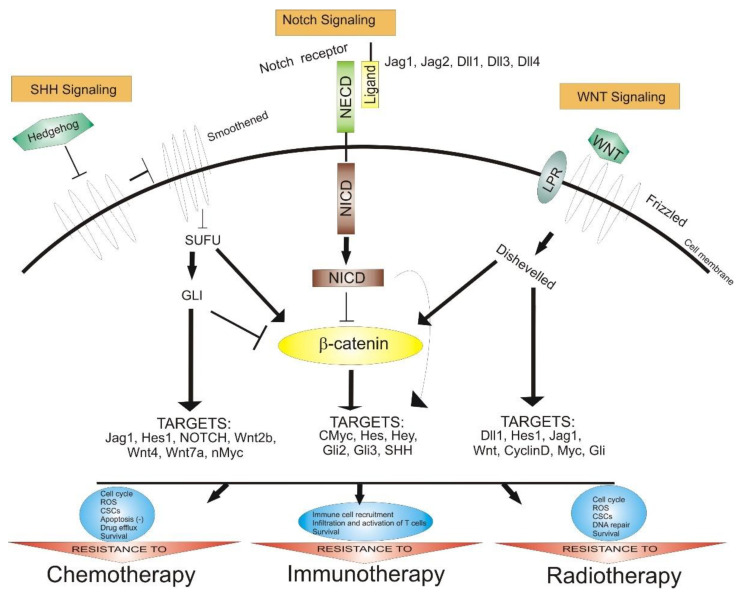
The role of Notch, SHH, and Wnt signaling pathways in anticancer therapy resistance. Abbreviations: SHH—Sonic hedgehog, SUFU—protein domain suppressor of fused protein, GLI—zinc finger protein, NECD—Notch extracellular domain, NICD—Notch intracellular domain, CSCs—cancer stem cells, ROS—reactive oxygen species, Jag—Notch ligand Jagged, Dll—delta-like ligand, Hes—hairy and enhancer of split, Myc—family of regulator genes and proto-oncogenes, point arrow—activation, block arrow—inhibition.

**Table 1 ijms-23-04602-t001:** Biomarkers of GSCs.

Marker	Category	Origin	Involved in	Reference
CD133/Prominin	Pantaspanglycoprotein family	Hematopoietic stem cells, endothelial progenitors, myogenic cells, and stem cells	Cell proliferation, migration, stem-cell-adjacent cell interactions	[45]
CD34	Transmembrane glycoprotein	Progenitor cells	Cell–cell adhesion, migration, hematopoietic stem cell attachment to the extracellular matrix	[46]
CD44	Glycoprotein	Stem cells	Adhesion in stem cell homing	[47,48]
CD15 (SSEA-1)	Trisaccharide	Developing neural stem cells and subventricular zone	Diagnosis as specific progenitor cell marker	[49,50]
Musashi-1	RNA-binding protein	Neural stem cells	Inhibiting the mRNAs’ translation	[51]
Nestin	Intermediate filament	Mammalian CNS stem cells during development	Tumor cell growth, metastasis, and GSCs’ self-renewal	[52]
SOX2 and HMG box	DNA-binding protein	Multipotent neural stem cells and embryonic stem cells	Sustaining neural and embryonic stem cell pluripotency	[53]
L1CAM (CD171)	Glycoprotein	Neural cells	Tumor growth, GSCs’ radiosensitivity, and DNA damage response regulation	[54]

**Table 2 ijms-23-04602-t002:** Targeted therapies in glioblastoma.

Target	Inhibitors	References
αvβ3 and αvβ5 integrin	Cilengitide	[72]
EGFR	Erlotinib, gefitinib, lapatinib, cetuximab, AEE788, EKB569, ZD6474	[73]
PDGFR	Imatinib mesylate, sorafenib, SU011248, PTK787	[74]
VEGFR	Sorafenib, valatanib, sunitinib, AEE788, AZD2171, ZD6474	[75]
mTOR	Temsirolimus, everolimus, sirolimus, AP23573	[76].
PKC	Tamoxifen, enzastaurin	[77].
Histone deacetylase	Depsipeptide, suberoylanilide hydroxamic acid	[78].
Farnesyltransferase	Lonafarnib, tipifarnib	[79]
Hsp90	17-AAG	[80]
Histone deacetylase	Depsipeptide, suberoylanilide hydroxamic acid	[81]
Proteasome	Bortezomib	[82]

## Data Availability

Not applicable.

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
