# Peer review of "Glioblastoma Stem Cells—Useful Tools in the Battle against Cancer"

_ijms, 2022, doi:10.3390/ijms23094602_

Round 1

Reviewer 1 Report

The manuscript represents a well-written literature review on stem cells in glioblastoma. As a comprehensible summary, it will be useful for scientists working in the given field.

Several interesting questions should be further highlighted in order to increase the scientific impact of the manuscript:

1) Authors have correctly noted the molecular classifications of glioblastoma as having predictive and/or prognostic role. However, in diagnostic surgical pathology immunohistochemistry is frequently considered a fast and reliable substitute for molecular genetic tests therefore it would be valuable to refer shortly to immunohistochemistry-based molecular classifications of glioblastoma. See, please,

  • Molecular classification of diffuse gliomas. Jakovlevs A, et al. Pol J Pathol. 2019;70(4):246-258. DOI: 10.5114/pjp.2019.93126.
  • A simplified integrated molecular and immunohistochemistry-based algorithm allows high accuracy prediction of glioblastoma transcriptional subtypes. Orzan F, et al. Lab Invest. 2020;100(10):1330-1344. DOI: 10.1038/s41374-020-0437-0.

2) To explain the radioresistance of glioma stem cells (noted in the Introduction and section N.4), please, include a discussion on replication stress as a resistance mechanism of glioblastoma stem cells (please, see, subsection 6.2 in Cancer Stem Cells: Significance in Origin, Pathogenesis and Treatment of Glioblastoma. Biserova K, et al. Cells. 2021;10(3):621. doi: 10.3390/cells10030621).

3) A significant fraction (63 out of 104, i.e., 61%) of the references are older than five years therefore inclusion of some more recent sources can be highly recommended. Please, feel free to use the advised sources or any other.

In conclusion, the manuscript is comprehensible and interesting therefore I recommend to accept it after the implementation of the suggested corrections.

Reviewer 2 Report

A brief summary:

The authors of this review discuss some concepts regarding the role of glioblastoma stem cells (GSCs) in GBM aggressiveness. After a general introduction with some GSCs biomarkers, the authors present the role of GSCs in the perivascular, perinecrotic or hypoxic and the invasive niches respectively. In conclusion they discuss molecular mechanisms with genetic and epigenetic changes considering Notch, Sonic hedgehog and Wnt signaling pathways.

Broad comment:

The review can certainly be of great interest as knowledge on GSCs is still inadequate and a better understanding of them could lead to interesting therapeutic perspectives. However, the work is not very thorough as the topics are described briefly and not in depth. The explanation of several concepts is incomplete because it is limited to basic knowledge without further investigation. All the concepts presented by the authors should be exhaustively elucidated. In the abstract the authors state that they describe the strategies for detecting and targeting GSCs. This information is not represented in the text. In conclusion, the review also lacks the future perspective and little or no reference is made about the current therapeutic approaches. At present, the work does not provide a significant contribution to the scientific community. An in-depth study of the topics can certainly improve the quality of the review.

Specific comments:

  1. I suggest deleting the term "multiforme", since it is obsolete today. It is no longer indicated in the most recent WHO classification. However, GBM abbreviation can be used for GlioBlastoMa. Please, check the reference below:

Louis DN, et al. The 2021 WHO Classification of Tumors of the Central Nervous System: a summary. Neuro Oncol. 2021 Aug 2;23(8):1231-1251. doi: 10.1093/neuonc/noab106. PMID: 34185076; PMCID: PMC8328013.

  1. I suggest initially describing the GSCs and TICs without mentioning the CSCs because it would be redundant. Authors should add differences between GSCs and TICs exposed by the scientific community.
  2. Lines 53-54: …”extensive DNA repair, quiescence, higher mitochondrial reserve” are presented as key features for GSC in the introduction, but they are not described in the following sections.
  3. Lines 56-58: Please, remove this sentence and add the most recent WHO classification of brain tumors (See the first point for the correct reference)
  4. Lines 61: I would suggest writing genomic signatures rather than changes.
  5. Lines 77-80: The authors describe this concept lightly. They should add more details on the role of the enzyme Aldehyde dehydrogenase -I and MGMT in mediating chemoresistance. Furthermore, they should add the reasons why this resistance is more pronounced in GSCs compared to GBM tumor cells.
  6. I suggest improving the quality of Table 1 formatting to make it more readable.
  7. Lines 95-100: I suggest being more specific with this concept since it represents a pivotal point for the discussion on GSCs. For instance, I suggest the following study regarding the evaluation of cellular differentiation in GBM and related findings:

Brooks LJ, Clements MP, Burden JJ, Kocher D, Richards L, Devesa SC, Zakka L, Woodberry M, Ellis M, Jaunmuktane Z, Brandner S, Morrison G, Pollard SM, Dirks PB, Marguerat S, Parrinello S. The white matter is a pro-differentiative niche for glioblastoma. Nat Commun. 2021 Apr 12;12(1):2184. doi: 10.1038/s41467-021-22225-w. PMID: 33846316; PMCID: PMC8042097.

  1. Section 3.2: The hypoxic microenvironment is a key characteristic for the growth and maintenance of GSCs. Nonetheless, the influence of hypoxia on GSCs is poorly described and not detailed. The authors should discuss more this point. I also suggest evaluating the following study, in which the authors described the role of hypoxia in radioresistance of GBM mediated by tyrosine kinase SRC as a key oncoprotein for a complex reorchestration of several signaling pathways and GBM invasion.

 Torrisi F, Vicario N, Spitale FM, Cammarata FP, Minafra L, Salvatorelli L, Russo G, Cuttone G, Valable S, Gulino R, Magro G, Parenti R. The Role of Hypoxia and SRC Tyrosine Kinase in Glioblastoma Invasiveness and Radioresistance. Cancers (Basel). 2020 Oct 4;12(10):2860. doi: 10.3390/cancers12102860. PMID: 33020459; PMCID: PMC7599682.

  1. Section 3.3: The cellular invasion and communication related to the SHH pathway meets a recent study that I suggest mentioning. In this study, the relationship between SHH and connexin43 (CX43) in GBM proliferation and migration was evaluated.

Torrisi F, Alberghina C, Lo Furno D, Zappalà A, Valable S, Li Volti G, Tibullo D, Vicario N, Parenti R. Connexin 43 and Sonic Hedgehog Pathway Interplay in Glioblastoma Cell Proliferation and Migration. Biology (Basel). 2021 Aug 12;10(8):767. doi: 10.3390/biology10080767. PMID: 34439999; PMCID: PMC8389699.

  1. Authors should add the reference in the text for Figure 1.
  2. The figure 1 does not provide the explanation of a complex mechanisms but it is simply a sort of outline about therapeutic approach for the treatment of GBM targeting GCSs. This concept could be described in a few words. Rather, I would suggest making a figure showing how some molecular features (i.e. Notch, Wnt, Sonic hedgehog signaling pathways) regulate the development of GSCs.
  3. Lines 287-289: This point needs to be explained better. Moreover, is it always true that hypomethylation leads to the oncogenes activation? Which targets are referred to the hypomethylation?
  4. Lines 284-303: This section does not appear to be related to WNT signaling. An independent section for epigenetic changes should be made.
  5. In Conclusions the authors mention again the CSCs. There is no evident discussion in the text describing the differences between CSCs and GSCs; therefore, I would suggest to delete CSCs.
  6. It is appropriate to perform a check of English grammar and to reformulate some sentences correctly. Some examples:

Lines 29-32: Check the grammar of this sentence.  “being” is not the correct tense.

Lines 33-36: The grammar structure is not correct; it is also not correct to start the sentence starting with "but" Check also dot repetition at the end.

Minor changes:

  1. Check the correct use of abbreviations. Sometimes words have already been abbreviated or not in the text. Examples below:

Line 15-16 Abstract: Check GSCs abbreviation

Line 137: Tumor-associated macrophages has already spelled.

Line 153: Glioblastoma should be abbreviated

Line 265: GSCs abbreviation.

Line 280: It has already been abbreviated.

Line 214: blood-brain barrier should be abbreviated.

Lines 63: typos for “the issue”, would be tissue?

Reviewer 3 Report

Dear authors,

You have done a careful and extensive review, but to point of view there are some aspects that should be clarified or corrected. These are the following:

  1. In the chapters, abstract and conclusions (lines 18 and 310), is spoken about metastases and GBM rarely metastasize.
  2. The term glioblastoma multiforme is not used and instead glioblastoma would be the correct term. (line 28).
  3. In lines 121-122 is stated, “Later, a variety of inhibitors became available, and their effectiveness was demonstrated on CNS tumors with in vitro and in vivo studies.” This must be explained with the corresponding drugs and references.
  4. In lines 126-128, the mechanisms of “vessels rupture” and “pericyte disintegration” should be explained.
  5. The mechanisms of how the immune function is suppressed should also be explained (line 130).
  6. In line 150 is written, “As VEGF and CSF, the CD34 role has been discussed in the diagnosis of various cancer pathologies.” As CSF means cerebrospinal fluid, this sentence should be better explained.
  7. CD34 is a membrane antigen. (line 154)
  8. (Line 188-189) “The close contact between the astrocytes, endothelial cells and pericytes facilitates the transport of ions and metabolites from the blood vessels to the brain.” Astrocytes, endothelial cells and pericytes are part of the brain. This sentence should be better explained.

Round 2

Reviewer 1 Report

Authors have responded to all previous comments and implemented extensive changes in the manuscript therefore I would suggest to accept the article for publication.

Reviewer 2 Report

The authors have significantly improved the quality of the manuscript. All the issues and criticalities of the previous version have been addressed and resolved.

Reviewer 3 Report

I agree with the corrections done.